# Vaccine-Associated Rubella Caused by the RA 27/3 Strain

**DOI:** 10.3390/vaccines11010065

**Published:** 2022-12-28

**Authors:** Manuel Paz, Magaly Padilla, Eduardo Perez, Jessica Sauceda, Adrian Camacho

**Affiliations:** 1Infectious Diseases Department, University Hospital “Dr. José Eleuterio Gonzalez”, Monterrey 64460, Mexico; 2Epidemiology Department, University Hospital “Dr. José Eleuterio Gonzalez”, Monterrey 64460, Mexico; 3Health Department of the State “Secretaría de Salud”, Monterrey 64000, Mexico

**Keywords:** rubella, vaccine, RA 27/3, adverse effect, vaccine-associated, Latin America, E1, Wistar

## Abstract

Vaccine-associated rubella is a very rare adverse effect after rubella vaccination; we report the characteristics of a young women who, after a vaccination campaign where she received three different vaccines against influenza, tetanus/diphtheria, and measles/rubella, developed a fever and rash consistent with rubella disease that was confirmed by sequencing of the virus. The evolution was favorable. The woman had two close contacts who did not develop the disease. Follow-up of the patient and her contacts was important to detect complications and for epidemiology surveillance.

## 1. Introduction

The introduction of universal vaccination against measles, rubella, and mumps is associated with a worldwide decrease in naturally transmitted cases [1]. 

The last naturally transmitted case of rubella reported in Mexico was in 2018, an imported case originating from China [2], however few cases of vaccine-associated rubella have ever been reported in Latin America. The rubella virus contains a single-stranded, positive-sense RNA genome [3]. It is the only member belonging to the genus Rubivirus of the family Matonaviridae, the genome is enveloped by a nucleocapsid and is surrounded by a lipid bilayer where two envelope glycoproteins, E1 and E2, are found. Rubella is propagated from person to person by expulsion of aerial droplets; furthermore, there are no known reservoirs in animals, making humans the only known hosts.

Most cases are mild and self-limiting infections; however, infections during the first trimester of pregnancy are the greatest concern as they can lead to severe congenital infections and significant morbidity [3]. After an incubation period of 14 to 21 days, rubella is characterized by a maculopapular rash that occurs initially on the face and spreads to the trunk and limbs. It self-limits and disappears within the first 48 h. Other nonspecific symptoms are a low-grade fever, malaise, transient arthralgias, and adenopathy, with the latter usually found in the posterior cervical and occipital region.

A diagnosis is made by obtaining serum and pharyngeal exudate samples for serological and molecular detection. The presence of specific IgM against rubella virus is present in serum samples in 50% of cases on the day of appearance of the rash; the optimal time for its detection is 5 days after the onset of symptoms [4]. Detection of the virus by a polymerase chain reaction (PCR) is reliable when samples are drawn between the first day of the rash up to 7 to 10 days later [5].

Rubella is a vaccine-preventable disease; owing to universal vaccination, a significant decrease in the circulation of the virus has been observed [6]. The most widely used rubella vaccine contains live attenuated viruses of the RA 27/3 strain, which grows in human diploid cell cultures. This vaccine is generally safe and well tolerated. Common adverse effects of this vaccine include fever, transient arthralgia, and adenopathies. To our knowledge, the frequency of inapparent infections is unknown; nevertheless, secretion of the live attenuated rubella virus may occur in the period from 7 to 28 days, but the development of clinical disease has only been reported in a few cases. Regarding the recipients with mild symptoms, most individuals may not seek medical attention. The most adverse event reported is fever (5–15%) and individuals often do not have other symptoms. Arthralgia is reported in up to 25% of adult women but it is not associated with other symptoms [5].

The rubella vaccine is contraindicated in severely immunocompromised patients, especially in those who are undergoing chemotherapy and long-term immunosuppressive therapy and individuals with HIV infections who are severely immunocompromised and have congenital immunodeficiency.

There are only five reported cases of vaccine-associated rubella in the literature. The first two cases were a young woman and a man from Singapore who received a measles, mumps, and rubella (MMR) vaccine (Priorix^®^, GlaxoSmithKline, Brentford, UK); two weeks later, vaccine-associated rubella was confirmed with the detection of the E1 gene region [6]. The third case was a 57-year-old man from Japan who received a live attenuated rubella vaccine (TO-336 strain) [7]. The fourth case found was fatal encephalitis associated with a measles–rubella vaccine in a 31-year-old man from Brazil who was previously healthy. Histopathology confirmed encephalitis and his immunochemistry was positive for the rubella virus. The fifth case we found was a female infant with severe combined immunodeficiency who presented with disseminated vaccine-acquired varicella and a vaccine-acquired rubella infection at 13 months of age. Vaccine-associated rubella is an extremely rare event. In this study, we describe the first case caused by the RA 27/3 strain reported in a patient from Latin America.

## 2. Case Presentation

A 40-year-old woman with a fever and rash was referred to the infection control unit at the University Hospital “Dr. José Eleuterio González” in Monterrey, Mexico. She had no history of recent travel or contact with sick persons or children under 5 years of age. She lived with a healthy 48-year-old man and an 82-year-old woman with no history of disease. She had a history of type 2 diabetes, hypertension, and obesity without pharmacological treatment. Fourteen days before the onset of symptoms she received vaccines against measles/rubella (the Edmonston–Zagreb strain for measles and the Wistar RA 27/3 strain for rubella), influenza, and tetanus/diphtheria at the same time during a community vaccination campaign.

Two days before our evaluation she began to have a maculo-papular rash on her face that progressed over the next 48 h to her chest, abdomen, and extremities (Figure 1). On the second day, a fever, headache, pharyngeal pain, and arthralgia in the elbows, wrists, and knees were noted. On physical examination, her body temperature was 38.7 °C; her blood pressure was 140/80 mm/Hg; her pulse rate was 96 beats/min; and her respiratory rate was14 resp/min. Submandibular adenopathy of 1 cm and a whiteish lesion with perilesional erythema in the oral cavity were noted. The rash was absent on the palms and soles.

Laboratory analysis showed a hemoglobin level of 13 g/dL, leucocytes of 4000 cells/μL, neutrophile of 2400 cells/μL, platelets of 196,000 cells/μL, aspartate aminotransferase of 27 U/L, alanine aminotransferase of 28 U/L, and negative heterophile antibodies. PCR-RT for zika, dengue, and chikungunya showed that they were not detected. The measles IgM was negative, the measles IgG was positive, the IgM Rubella was indeterminate, and the IgG was negative. The pharyngeal swab RT-PCR showed that measles was not detected, and that rubella was detected (CT = 38.4). The rubella virus sequence obtained from the patient corresponded to the region of the E1 gene of the Wistar RA 27/3 strain of the vaccine, so the case was determined as vaccine-associated rubella and isolation was indicated. During the next 3 days, she presented with arthralgia and a fever, after which her clinical symptoms dissipated without complications and the rash disappeared within a week.

The patient and two contacts remained in isolation and under surveillance for 10 days without complications or symptoms. Once the suspicion of feverish exanthematic disease began, the first measure was to isolate the patient and find her contacts to place them in quarantine while they were monitored for the appearance of symptoms that could raise suspicions that they were also infected. In addition, the batch of vaccines that were used during the vaccination campaign was monitored in a search for any abnormalities and we monitored any other suspected cases in the region.

First, based on the current epidemiology, knowing that the last case of rubella in Mexico was reported in 2018, the probability of this case being caused by a wild rubella virus is very low. Because of the history of recent immunization with three different vaccines in which the rubella vaccine was included, the development of a fever and rash 2 weeks after vaccination made us consider either measles or rubella. After the PCR for measles was negative, this disease was excluded; the positive measles IgG inferred previous exposure. The negative serology for rubella and the identification of the E1 gene of the RA 27/3 strain in the pharyngeal swab confirmed the presence of the virus. The fact that no new cases occurred during the follow-up of the patient made us think that this was vaccine-associated rubella.

## 3. Discussion

We present the case of a patient who developed rubella after vaccination.

Momoka et al. [7] reported the case of a patient in Japan with a history of type 2 diabetes and high blood pressure who received a rubella vaccine 16 days before the onset of symptoms. Unlike our case, in which there were no circulating cases of rubella at the time of diagnosis, at the time of vaccination in the case of Momoka et al., there were 47 cases of rubella reported in the previous 8 months; in the study of this case, a PCR for rubella with genotype 1 was carried out which identified the vaccine strain.

Sean Wei et al. [6] described two cases of adults presenting with post-vaccination rubella disease. One of the cases was a 61-year-old from Singapore who received a dose of the MMR vaccine along with a dose of the inactivated quadrivalent influenza vaccine; genotyping later confirmed the association with the vaccine strain. The other case was a 31-year-old woman who developed rubella after rubella vaccination. Both patients had the typical symptoms of rubella and a benign clinical course, and they recovered without complications.

Other previous cases include a pediatric patient with severe combined immunodeficiency who developed rubella and chickenpox associated with vaccination [8] and for a Brazilian man who presented with fatal encephalitis following rubella vaccination, it was confirmed that the viral isolation matched the vaccine strain [9].

The current case, as in one of the reported cases, was also vaccinated against influenza; it is unknown if this could have any causative effect on the development of clinical disease. The diagnosis of this case was made by carrying out a PCR for rubella in pharyngeal exudate with detection of the E1 gene, which coincides with the RA 27/3 strain which was used for rubella vaccination in Mexico at that time. Identification of the E1 gene by RT-PCR in the patient corresponds with the Wistar RA 27/3 strain; this E1 protein is used in the Wistar vaccine to induce an immune response. The fact that the E1 gene was detected in the context of a recently vaccinated patient who developed symptoms of rubella with no cases of the disease reported in the region demonstrates that, like in other cases, this corresponds to the identification of a live attenuated virus.

It drew our attention that this patient received other vaccines at the same time. To our knowledge, we found no information about the risk of developing infections associated with vaccination with this history. Although immunocompromised patients are at a high risk of post-vaccination complications, we do not have information on whether immunosuppression associated with uncontrolled diabetes plays a role in the development of rubella associated with vaccination, since this immunosuppression is different from that observed in other patients associated with CD4 T cell deficiency. Therefore, the cause of these complications may be associated with a multifactorial situation, which has not yet been established.

Unlike the wild rubella strain, the rubella vaccine strain is not as infectious. Isolation precautions are important in naturally occurring rubella infections due to the wild strain; however, there is no information about isolation in patients with infections caused by vaccine strains. Only one study was found where the potential transmission of the virus associated with vaccination was described; however, there was no genomic detection in that study [10].

Because the last case of rubella imported into Mexico was in 2018 [11], the identification of rubella by a wild virus or associated with vaccination is important. The risk factors for the development of rubella associated with vaccination are not known. The incidence is unknown; patients have mild symptoms and may not seek medical attention. The recommendations for epidemiological surveillance and the control of infections are established;however, there is controversy about the contagion capacity in vaccine cases.

## 4. Conclusions

In conclusion, vaccine-associated rubella is an uncommon event and is classified as a complication after vaccination. It should be confirmed whether it is an infection caused by a wild or vaccinal virus. The risk factors for this outcome are not well established and may differ between different cases. Although the risk of transmission to other people is not as high as in infections associated with wild viruses, the handling of contacts and the patient must be approached with the same precautions.

## Figures and Tables

**Figure 1 vaccines-11-00065-f001:**
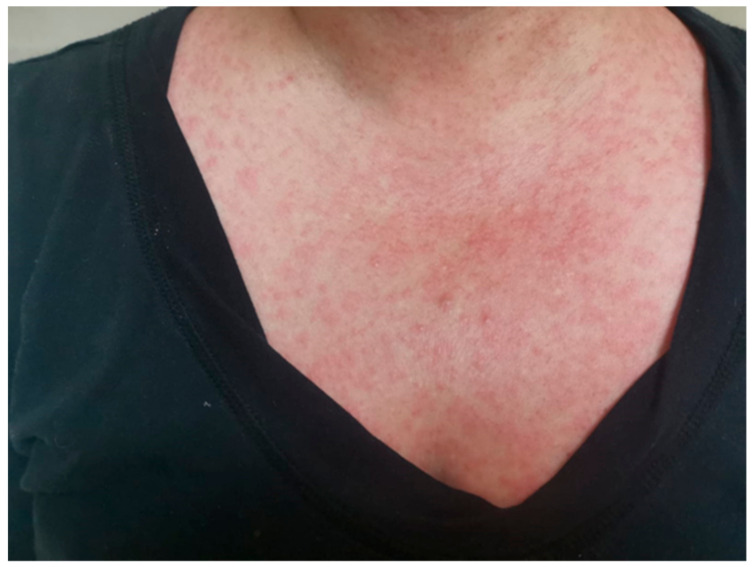
Maculo-papular rash.

## Data Availability

Not applicable.

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
