# Peer review of "Vaccine-Associated Rubella Caused by the RA 27/3 Strain"

_vaccines, 2022, doi:10.3390/vaccines11010065_

Round 1

Reviewer 1 Report

Although this case report entitled “Vaccine-associated Rubella caused by RA 27/3 strain” is informative, there are several concerns that need to be addressed as follows.

1.    The frequency of inapparent infections or recipients with mild symptoms after rubella vaccination.

2.    The identification of rubella by wild virus or live attenuated virus.

3.    More detailed information on the immunosuppressive state of the patients.

Finally, the authors should summarize case presentation in one table.

Author Response

RESPONSE TO REVIEWER 1

Point 1: “The frequency of inapparent infections or recipients with mild symptoms after rubella vaccination.”

Response 1: We thank the reviewer´s observation. To our knowledge, the frequency of inapparent infections is unknown, nevertheless, secretion of lived attenuated rubella virus may occur in the period from seven to 28 days but the development of clinical disease has just been reported in a few cases. Regarding the recipients with mild symptoms, most of the patients may not seek medical attention. Most adverse events reported are fever (5%-15%) and do not have other symptoms. Arthralgias are reported in up to 25% of adult women but is not associated with other symptoms.

Point 2: “The identification of rubella by wild virus or live attenuated virus.”

Response 2: Identification of the E1 gene by RT-PCR in the patient corresponds with the Wistar RA 27/3 strain, this E1 protein is used in the Wistar vaccine to induce an immune response. The fact that the E1 gene was detected in the context of a recently vaccinated patient that developed symptoms of rubella with no cases reported of the disease in the region demonstrates, like in other cases that this corresponds to an identification of a live attenuated virus.

Point 3: “More detailed information on the immunosuppressive state of the patients.”

Response 3: Rubella vaccine is contraindicated in severely immunocompromised patients, especially in those who are in chemotherapy, long-term immunosuppressive therapy, and HIV infections severely immunocompromised and congenital immunodeficiency. To our knowledge, nine patients with severe combined immunodeficiency, have been reported by Patel et all, with vaccine-associated rotaviral diarrhea. Only one case of Disseminated vaccine-strain rubella (RA27/3) in an immunocompromised patient has been reported. There is no information about the risk in diabetic patients regarding rubella vaccine-related 

Point 4: “case presentation in one table”

Travel

No recent travel

Personal history

40-year-old woman, Type 2 diabetes, hypertension, obesity

Recent Vaccines

Measles/Rubella (Edmonstos-Zagreb Strain for measles and Wistar RA 27/3 Strain for rubella), Influenza, and Tetanus/diphtheria.

Clinical features

14 days after vaccination, she had maculo-papular rash, fever and arthralgias.

Test result

Measles virus IgM

Measles virus IgG

Rubella virus IgM

Rubella virus IgG

Measles pharyngeal swab RT-PCR

Rubella pharyngeal swab RT-PCR

Negative

Positive

Indeterminate

Negative

Negative

Positive

Reviewer 2 Report

Comments are included in the manuscript

Author Response

RESPONSE TO REVIEWER 2

Point 1: if there are five previous cases and you describe them, why only the first three cases?, what is about. the other two cases?

Response 1: We thank the reviewers observation. Here we describe the other two cases from line 50 to line 54. The fourth case found, was a fatal encephalitis associated with measles-rubella vaccine in a 31-year-old man, previously health from Brazil. Histopathology confirmed encephalitis and immunochemistry was positive for rubella virus. The fifth case we found was a female infant with severe combined immunodeficiency who presented with disseminated vaccine-acquired varicella and vaccine-acquired rubella infection at 13 months age.

Point 2: Measles serological result positive and virus negative  and Rubella serological result negative and virus positive and clinical and epidemiological information: I think you must explain the the final diagnostic and the way on which you arrived to it, in my opinnion it will be the most interesting part of the manuscript.

Response 2: First, based on the current epidemiology, knowing that the last case of rubella in Mexico was reported in 2018, the probability of this case being caused by a wild rubella virus is very low. Because of the history of recent immunization with 3 different vac-cines in which the rubella vaccine was included, the development of fever and rash 2 weeks after vaccination made us think between measles and rubella. After the PCR for measles was negative this disease was excluded, the measles IgG positive talk about a previous exposure. The negative serology for rubella and the identification of the E1 gene of the RA 27/3 strain at the pharyngeal swab confirm the presence of the virus. The fact that no new cases have occurred during the follow-up of the patient, made us think that this is a vaccine-associated rubella.

Point 3: The manuscript can be improved if you include all tha epidemiological study of the case (you can explain it and the way on which you worked with historical data of the patient and her environment) and according to it and the clinical and laboratory analysis it could bee interesting to present your hipothesis (risk factors, vaccine-case links, role of other vaccines…….)

Response 3: It drew our attention that this patient received other vaccines at the same time. To our knowledge, we found no information about the risk of developing infections associated with vaccination with this history. Although immunocompromised patients are at high risk of post-vaccination complications, we do not have information on whether immunosuppression associated with uncontrolled diabetes plays a role in the development of rubella associated with vaccination, since this immunosuppression is different from that observed in other patients associated with CD4 T cell deficiency. Therefore, the cause of these complications may be associated with a multifactorial situation, which has not yet been established.

Point 4: The manuscirpt needs to include all the case study and references or comments related to it because it seems only a small reference in order to say that it is the case number 5 and a samll comment on the first 3 cases.

Response 4: The complete cases were included in the point 1, also at the discussion this cases that were missed are already included in line 107 to line 122. Please let me know if more information need to be added

Point 5: Also conclusion is a general idea about rubella vaccination without anyone reference to the case and its importance

Response 5: In conclusion, vaccine-associated rubella is an uncommon event and is classified as a complication after vaccination. It should be confirmed whether it is an infection by a wild or vaccinal virus. The risk factors for this outcome are not well established and may differ between different cases. Although the risk of transmission to other people is not as high as in infections associated with wild viruses, the handling of contacts and the patient must be taken with the same precautions.

Reviewer 3 Report

Estimated Authors,

thank you for your interesting case report. In this short study, Authors have reported on a case of post-vaccine rubella infection, with distinctive features that are properly summarized by Authors. Potential interest, limits of interpretation, and significance for readers of Vaccines are discussed in both introduction and discussion sections. Appropriate references are included and discussed.

In summary, from the point of view of the present reviewer, the study is properly written, and could be accepted as it is.

Author Response

We thank the reviewer’s observation.

Round 2

Reviewer 1 Report

The paper is now suitable for publication.

Reviewer 2 Report

 The manuscript has been imporved according to the preliminary comments